# Qualitative Classification of Proximal Femoral Bone Using Geometric Features and Texture Analysis in Collected MRI Images for Bone Density Evaluation

**DOI:** 10.3390/s23177612

**Published:** 2023-09-02

**Authors:** Mojtaba Najafi, Tohid Yousefi Rezaii, Sebelan Danishvar, Seyed Naser Razavi

**Affiliations:** 1Faculty of Electrical and Computer Engineering, University of Tabriz, Tabriz 51666-16471, Iran; mojtaba.najafi@tabrizu.ac.ir (M.N.); yousefit@tabrizu.ac.ir (T.Y.R.); razaviii@tabrizu.ac.ir (S.N.R.); 2College of Engineering, Design and Physical Sciences, Brunel University London, Uxbridge UB8 3PH, UK

**Keywords:** osteoporosis, magnetic resonance imaging, dual energy X-ray absorptiometry, machine learning

## Abstract

The aim of this study was to use geometric features and texture analysis to discriminate between healthy and unhealthy femurs and to identify the most influential features. We scanned proximal femoral bone (PFB) of 284 Iranian cases (21 to 83 years old) using different dual-energy X-ray absorptiometry (DEXA) scanners and magnetic resonance imaging (MRI) machines. Subjects were labeled as “healthy” (T-score > −0.9) and “unhealthy” based on the results of DEXA scans. Based on the geometry and texture of the PFB in MRI, 204 features were retrieved. We used support vector machine (SVM) with different kernels, decision tree, and logistic regression algorithms as classifiers and the Genetic algorithm (GA) to select the best set of features and to maximize accuracy. There were 185 participants classified as healthy and 99 as unhealthy. The SVM with radial basis function kernels had the best performance (89.08%) and the most influential features were geometrical ones. Even though our findings show the high performance of this model, further investigation with more subjects is suggested. To our knowledge, this is the first study that investigates qualitative classification of PFBs based on MRI with reference to DEXA scans using machine learning methods and the GA.

## 1. Introduction

Osteoporosis is a systemic skeletal disease in which bone fragility and risk of fracture increase with loss of bone density and microarchitectural deterioration of bone tissue. Many parameters such as age, race, gender, hormonal disorders, heredity, and poor dietary habits correlate with the progression of the disease [1]. Osteoporosis can be considered to be a result of a lack of calcium as well as other minerals in the bones. This disease can be classified into the following three categories: primary osteoporosis including idiopathic osteoporosis, type 1 osteoporosis due to the estrogen deficiency common in postmenopausal women, and type 2 osteoporosis due to an estrogen deficiency with a loss of bone density. Vertebral fractures can be considered to be one of the common consequences of osteoporosis, which leads to constant pain, skeletal–muscular abnormalities, respiratory abnormalities, and deterioration of physical performance and quality of life. Also, there will be a chance of sustaining a fracture after an initial fracture. Physiotherapy can be considered a special non-pharmacological method that can be useful and effective for patients who have osteoporosis, through manual methods, resistance exercises, etc. [2,3].

Osteoporosis is common, preventable, and treatable [4,5]. However, it is a silent disease as it may be asymptomatic for many years until the fracture occurs, which can be physically debilitating and consequently cause a financial burden on the patient [6,7]. Without prevention and screening, the osteoporosis-related morbidity and mortality will greatly burden healthcare systems throughout the world [8,9]. Thus, it is key to diagnose and treat osteoporosis before a fracture occurs.

Bone mineral density (BMD) is currently the main factor for determining bone quality and risk of osteoporosis [10]. Dual-energy X-ray absorptiometry (DEXA) is considered the clinical gold standard for BMD evaluation. According to the guidelines of the National Osteoporosis Foundation, the indications for DEXA examination are: (a) women 65 years old and men 70 years and older, (b) postmenopausal women and men over 50 who have suffered bone fractures, and (c) adults who suffer from rheumatoid arthritis or use medications such as glucocorticoids and have low bone mass [2,3,11]. However, DEXA scans do not give any information regarding the bone microarchitecture and mechanical properties, which are also crucial factors in bone resistance to fracture [12,13]. To obtain these factors, other non-invasive methods such as conventional radiology, computed tomography, magnetic resonance imaging (MRI), and ultrasound have been investigated [14,15].

MRI is a common imaging technique used in imaging centers. The feasibility of structural analysis of proximal femoral bone (PFB), one of the most clinically important sites of fracture, using MRI has been established [16,17]. Use of quantitative MRI techniques in assessing osteoporosis and fracture risk in the PFB has been documented [18,19]. Image segmentation and quantification to analyze osteoporosis and bone fracture risk using these methods are time-consuming and impractical [20]. The use of machine learning techniques (deep learning) such as convolutional networks [21] and deep convolutional neural networks [22] has shown promising results in solving segmentation problems.

If we can determine the bone quality from MRI, when patients are referred to imaging centers to check the health of body organs such as the bladder, intestine, liver, kidneys, ovaries, and breasts in women and the prostate in men, we can detect many patients with osteoporosis or who are on the verge of developing osteoporosis earlier and use appropriate treatments to reduce the number of patients in society. However, different acquisition protocols influence the results of segmentation [23,24] and texture analysis [25].

The aim of our study is to investigate whether the discrimination of unhealthy femurs using images with different scan parameters (and from different hospitals) is practical or not. Another goal in our study is to identify important features in the classification of healthy and unhealthy femurs.

To our knowledge, this is the first study that uses machine learning and a genetic selection algorithm to evaluate proximal femoral bone quality based on MRI with reference to the BMDs from DEXA scans. We used machine learning methods to qualitatively classify proximal femoral bones into two classes (healthy and unhealthy bones (bones with osteopenia or osteoporosis)) using geometric features and texture analysis in MRI. Finally, we verified our classification results against a classification based on the results of DEXA scans as our reference.

The main contributions of this article are as follows:The compilation of a comprehensive MRI database available for the diagnosis of osteoporosis;The provision of a smart algorithm based on machine learning techniques to evaluate bone density;Achievement of the highest detection speed for bone density assessment;A comparison of the proposed method with other deep learning methods.

## 2. Methods

In this section, the collection of MRI images, pre-processing operations, data segmentation, classifications, and features used are fully described.

### 2.1. Data Collection

Our collected data included 284 images taken from 21 to 83-year-old Iranians with no history of bone fracture (135 men, average age of 56, standard deviation of 13, and with an average BMI of 27.8 and 149 women, average age of 49.5, standard deviation of 16, and with an average BMI of 24.3). The data were collected from the Ronash imaging center in Dezful city, Allameh Bohlul hospital in Gonabad city, and Noh-e-Dey hospital in Torbat-e-Heydarieh city in Iran. We obtained the informed consent of all subjects in writing prior to the tests and the tests were approved by the Ethics Committee of the hospital (with license number IRB.1398.9.26). In general, the following patients were excluded from the testing process: patients with a lack of personal consent for testing, people who used a pacemaker, and people who had an implant or a metal foreign body in their body (such as having a piece of metal in the brain or eye or platinum in bone).

Subjects were referred by a physician to imaging centers to be scanned from the pelvic region (and in most cases other regions as well). Scan parameters were set by specialists according to their experience. All three centers used 1.5T MRI machines (two had GE Optima 450 MRI machines and the other had a Siemens MAGENTOM Avanto MRI machine). These scan parameters are presented below:T1/TR = 415.00 ms/TE = 19.00 ms/Flip-Angle = 150 (44 referred, 33 healthy, 11 unhealthy)T1/TR = 536.00 ms/TE = 11.00 ms/Flip-Angle = 180 (43 referred, 31 healthy, 12 unhealthy)T1/TR = 4070.00 ms/TE = 33.00 ms/Flip-Angle = 180 (42 referred, 29 healthy, 13 unhealthy)T1/TR = 420.00 ms/TE = 22.00 ms/Flip-Angle = 180 (45 referred, 36 healthy, 9 unhealthy)T2/TR = 3600.00 ms/TE = 80.00 ms/Flip-Angle = 150 (45 referred, 19 healthy, 26 unhealthy)T2/TR = 7840.00 ms/TE = 109.00 ms/Flip-Angle = 150 (41 referred, 28 healthy, 13 unhealthy)T1/TR = 389.00 ms/TE = 13.42 ms/Flip-Angle = 110 (24 referred, 9 healthy, 15 unhealthy)

The BMDs of the subjects were obtained using three DEXA scanners (NORLAND bone densitometer). The scan parameters were set by the scanner for the femur automatically and not by the operator.

The person who recorded the MRI images had an MRI fellowship (specialized in radiology) and the person who recorded the DEXA scans was a radiologist. The time it took to obtain MRI images of the femur was approximately 15 min, and the time it took to obtain the BMD of the femur using the DEXA technique was approximately 2 min.

### 2.2. Programming Environment and Settings

Reading of the MRI images (saved in DICOM format), their conversion into usable numeric arrays, image segmentation, and feature extraction were implemented in Python and in the Anaconda-Spider environment. We used MATLAB R2021b to implement classification and feature selection processes.

We have made our program available to the public on https://github.com (file name: Mojtab2023/Classification-of-Proximal-Femoral-Bone-Using-Geometric-Features-and-Texture-Analysis-in-MR-Images-f (1 January 2022)).

### 2.3. Overall Procedure

Figure 1 shows the flowchart of the procedure we used to achieve classification of the PFB. We used a 10-fold cross-validation technique to carry out our experiment. To this end, the dataset was divided into 10 non-overlapping sections. The remaining section was used for testing after the nine sections were used for training. The process was repeated 10 times until all sections were taken into account in the testing procedure.

The PFBs were divided into the two groups of unhealthy (with the risk of osteopenia or osteoporosis; T score ≤ −0.9) and healthy (T score > −0.9) based on the result of the DEXA scans and labeled 1 and 0, respectively.

Because the number of images associated with the healthy label exceeded the number of images associated with the unhealthy label, which may cause the majority class to be considered in the classification, the number of minority classes was increased. We artificially increased the data of the minority class using a data augmentation technique. In this way, the rotation range of the images was considered to be 40 degrees. Also, the horizontal shift of the images and the vertical shift of the images was considered to be 0.2%.

### 2.4. Segmentation

We used a 5 × 5 pixel median filter to reduce salt and pepper noise in the MRI. Then, segmentation was performed using the region growing method or the active contour method.

In the region growing method, we used the Canny edge detection technique. The starting point for the edge detection algorithm is important and affects the result. In order to find the best starting point, we used 2 different methods. One method was using the geometric features of PFB and the Hough transform to obtain the center of the Hough circle as the starting point and the other was choosing a pixel inside the bone with the highest intensity (where the image’s histogram in the horizontal and vertical directions was the maximum). We evaluated the two methods used based on two indices (the Dice Similarity Coefficient (DSC) [26] and the Jacard Index (also called Intersection over Union, IoU)) [27] to determine the efficiency of each method. These two methods can be defined as Equations (1) and (2), where *S* and *T*, correspondingly, represent the segmentation result and the ground truth. Figure 2 shows a sample of the segmented results by both techniques. The best result from these two methods was chosen for feature extraction. In thirteen cases, segmentation failed by both methods and the boundary of the femur was determined manually. Four out of these thirteen images had scan parameters of (T1/TR = 389.00 ms/TE = 13.42 ms/Flip-Angle = 110).
(1)DSC=2|S∩T||S|+|T|
(2)IoU=|S∩T||S∪T|

After segmentation, features were extracted. Figure 3 shows the basic geometrical features that were used in this study. The parameters W1, W2, etc. are defined as geometric features in Figure 3. The purpose of this definition of these geometric parameters that are marked on the figure was to find out whether there is a relationship between the defined geometric parameters with osteoporosis. To put it more simply, parameters W1, W2, etc. were defined by ourselves as geometric features and are not shorthand for a specific phrase or word.

We created an averaged image from unhealthy (based on DEXA results) PFBs and another averaged image from healthy PFBs in our training dataset after flipping (for right femurs), cropping, and framing PFB images at similar positions. For images with the same scan parameters, the difference between these two images indicated the important regions in which these two classes differ the most, i.e., the boundaries of the PFB. This means that characteristics of the boundaries may represent the difference between the two PFB classes. When we used all images with different scan parameters, the inside region also became more apparent in the subtracted image (see Figure 4).

Five descriptive curves were also extracted from the cropped image of the bone. These curves were named C1-0, C1-1, C1-2, C2-0, and C4-0 (see Figure 5).

Each curve was obtained by measuring the distance of the bone’s boundaries from the edge of the image (in the direction indicated by the arrow) along the marked regions. These curves describe the characteristics of the bone boundaries. Figure 6 shows two of the rustled curves for a PFB as an example.

Then, statistical functions such as the minimum, maximum, mean, variance, elongation, skewness, and first and second moments of the signal (moment1 and moment2 in Table 1) of these descriptors were calculated. We defined some features as below:(3)Average ramp=Δy/Δx
(4)$$\mathrm{First}\;\mathrm{to}\;\mathrm{last}\;\mathrm{point}\;\mathrm{ramp}=\Delta y_{1}/\Delta x_{1}$$

Figure 7 shows how these deltas were defined. Furthermore, we defined another feature to remove the effect of slope from our curves (to have more normal-distribution-like characteristics) to make functions such as skewness and kurtosis meaningful. This feature was defined as below:(5)Ramp removed sig=signal−averageramp

Features such as the pixel gray-scale average (as an indicator of bone marrow fat and the BMD) [28], the fractal dimension (using the Boxing count method), and the gray-level co-occurrence matrix were also derived from texture analysis (for further information on the image segmentation and texture analysis refer to the work of Larroza et al. [29]). A total of 68 primary features were extracted. We created a feature vector (FV) with these features as well as their 2nd and 3rd powers, i.e., a total of 204 features. Next, we used decision tree, logistic regression, and support vector machine (SVM) algorithms to classify the data.

### 2.5. Classification Algorithms

In this section, the classifications used and compared in this study are reviewed.

#### 2.5.1. Classification Using Decision Tree

Decision tree (DT) is a versatile nonlinear machine learning model composed of hierarchical decision rules established during the training procedure to perform a classification task by recursively splitting independent factors into different groups. In [30,31,32], DT was used to perform disease diagnoses. Some studies also have employed and compared DT with other machine learning algorithms [33]. Here, we utilized DT as well as other classifiers to evaluate the extracted features.

#### 2.5.2. Classification Using Logistic Regression

Logistic regression is a simple linear supervised binary classifier. Some studies have used logistic regression for medical purposes [34,35,36]. This classifier was employed in our experiments as a benchmark because, in recent years, it has been widely used in studies related to the classification of different types of osteoporosis.

#### 2.5.3. Classification Based on the SVM Algorithm

SVM is one of the most-used algorithms for texture analyses of MRI [37]. It is computationally cheaper than a neural network and can produce similar or even better results [38].

Polynomial, linear, radial basis function (RBF), and Gaussian kernels were used in the implementation of the SVM algorithm in our study.

### 2.6. Feature Selection Using the Genetic Algorithm

Feature selection is necessary to find the most relevant features to improve the classification performance due to the variable discrimination power of the extracted features. The classification performance is hampered by redundant or irrelevant features; when more features are added to a model, the classification performance suffers [39].

The Genetic algorithm (GA) has been successfully used for feature selection in MRI segmentation and classification and also in combination with the SVM [40,41].

In the GA, the total population was set to 100, with a maximum of 500 iterations. The flowchart of this procedure is shown in Figure 1b. In our study, we used a cost function that would result in the best accuracy with the least number of features (*NOF*) as defined below:(6)$$\mathrm{Cos}t\;(\mathrm{FV})={\left(Accuracy+1/NOF\right)}^{-1}$$

## 3. Results

Based on the results of their DEXA scans and T scores ≥ −0.9, 185 subjects were labeled as “healthy”, while 99 subjects were labeled as “unhealthy”.

As the first part of our study, we employed SVM with different kernels (SVM-linear, SVM-Polynomial, SVM-RBF, and SVM-Gaussian), decision tree, and logistic regression classifiers to classify healthy and unhealthy cases using all 204 features. All the experiments were performed using the 10-fold cross-validation technique. Given the inherently random characteristic of the classifier training procedure and the fact that data partitioning impacts the final result, we repeated the experiments ten times. The results are shown in Figure 8.

As we can see, with all 204 features, the highest accuracies were obtained from the SVM with RBF and Gaussian kernels (more than 80%) and the lowest accuracy was from the logistic regression (above 60%).

Figure 9 shows how the classification performance of different algorithms improved in 500 iterations of the GA. When we selected features based on the GA, the accuracy of the SVM-RBF increased from about 83% to 89% using 56 features, while the accuracy of SVM-Gaussian, using only 7 features, increased from about 81% to 86%. Reducing the number of features improved the classification performance and increased the computational speed as was expected based on the literature [40,41].

Table 2 shows the best results (using a confusion matrix) achieved based on the features selected by the GA for each classifier and its corresponding number of selected features. According to Table 2, as can be seen, the GA has selected different features and the number of features for different categories. For example, 60 features, 56 features, and 7 features are selected for SVM-Linear, SVM-RBF, and SVM-Gaussian, respectively. The types of features selected by the GA are discussed below.

In SVM-Gaussian, the seven selected features were W0, C1-1 (max. of second derivative), C1-1 (variance) 2, C2-0 (first to last point ramp) 2, C4-0 (moment2 (ramp_removed_sig)) 2, and fractal dimensions. On the other hand, the main selected features for SVM-RBF were based on geometric features (15 features) and C2-0 (9 features). The most influential features in our study in descending order were geometrical features (in Table 1), C1-2, C2-0, fractal dimensions, C1-0, C1-1, and, finally, features based on the texture. The most-used features across all classifiers were y1, W2, and C1-1 (kurtosis (ramp_removed_sig)).

Although every primary feature was used either in its first, second, or third order, a total of 105 features were not used by any classifier, half of which belonged to the third power of the primary features. This was expected, as we designed the cost function of the GA not only to improve the classification performance but also to reduce the dimensionality of the feature vector.

Each run of the GA may result in a different set of features. Figure 10 shows the selection process of the GA for SVM-RBF, SVM-Polynomial, SVM-Gaussian, and decision tree during the 500 iterations. Each of these binary images has 500 columns (corresponding to 500 iterations) and 204 rows (associated with 204 features). Ones (white pixels) and zeros (black pixels) represent selected or missed corresponding features in the relevant iteration. The horizontal white line represents a feature that was steadily chosen by the GA. The most unstable accuracy based on feature selection by the GA was for the decision tree model with less than 2% fluctuations in accuracy.

In another experiment, different classifiers were employed to evaluate the feature vector that resulted in the best classification performance by the SVM-RBF classifier. All experiments were performed 10 times and Figure 11 shows the results.

Comparing the results obtained by all features and the 56 features selected by the GA for the SVM-RBF, which are shown in Figure 8 and Figure 11, respectively, reveals that the selected features outperform the result of using all features in terms of classification accuracy. However, the accuracy is still considerably lower than that of the GA-selected features.

In other words, the best performance of each classifier was achieved by the feature vector optimized by the GA for that classifier. Another notable finding from Figure 11 is the small variance in the SVM-RBF with the optimized feature vector. Since the feature vector is optimized for this specific classifier, the results of the 10 experiments are close. This may be considered to be a stability factor for the selected feature vector.

Among the different scan parameters used in our study, images with scan parameters of T1/TR = 536.00 ms/TE = 11.00 ms/Flip-Angle = 180 produced the best results with an accuracy of 97.73% (only one false unhealthy detection).

The reason for our use of hand-crafted feature selection/extraction techniques and manual classification in this work is due to the low number of images obtained. Due to the fact that the database of recent studies is not the same, it does not seem fair to evaluate the studies one by one. However, we compared our proposed model with some well-known deep convolutional networks based on our collected database. The compared convolutional networks included ResNet17 [42], VGG19 [43], GoogleNet [44], and Inceptionv3 [45]. The results obtained after 500 iterations are presented in Table 3. As is shown, the obtained classification accuracy for the ResNet and VGG networks is almost equal to the proposed model. However, these networks need more time to reach convergence, which will increase the computational load of the algorithm. The performance of convolutional networks can be increased for the purpose of classification in the case of increasing data. According to Table 3, it has been proven that, for data with smaller dimensions, the proposed method has a higher performance in terms of speed and accuracy.

Due to the nature of MRI, the slightest movement of the patient can cause noise in the captured images. Accordingly, the proposed method should be designed in such a way that it can be resistant to movement and environmental noises. In order to evaluate the proposed model more accurately, we artificially added Gaussian white noise in a wide range of different SNRs to the images and present the results based on the compared ResNet17, VGG19, GoogleNet, and Inceptionv3 pre-trained deep networks in Figure 12. According to Figure 12, as can be seen, the proposed model based on the selection/extraction of engineering features can be more resistant to noise than the ResNet17, VGG19, GoogleNet, and Inceptionv3 networks in a wide range of different SNRs. So, at an SNR = −4, it still remains above 80%.

## 4. Discussion

Studies have shown that MRI scans and extracted parameters can be used to discriminate between healthy and unhealthy human bones quantitatively and to assess fracture risk. Deniz et al. [20] used deep convolutional neural networks (CNNs) to carry out segmentation of the PFBs using MRI automatically and compared their results with PFBs manually segmented by an expert. They used 86 subjects and their segmentation method resulted in 95% accuracy. This segmentation method may produce better results than our classification method; however, it needs a higher number of subjects and the same number of scan parameters.

Our results demonstrate the superiority of the SVM algorithm. However, we used a basic function for decision tree and logistic regression, while more sophisticated models for these techniques may produce better results. In addition, we investigated the effect of extracting geometric features (see Figure 2) on osteoporosis for the first time among previous studies, and we concluded that there is a direct relationship between geometric features and the early detection of osteoporosis. In addition, we compared our proposed method, which is based on extracting engineering features from images, to deep learning methods and concluded that our proposed method is resistant to environmental noises and can be used in practice.

Makond et al. [32] used a decision tree model based on CHAID’s algorithms for postmenopausal osteoporosis screening, which resulted in a sensitivity (TP/(TP + FN)) of 92.3% and a 82.8% positive predictive value (TP/(TP + FP)). Ferizi et al. [46] used different classifiers to predict fragility fractures from MRI data. The top three classifiers in their study were RUS-boosted trees, logistic regression, and linear discriminant, which outperformed linear, cubic, and quadratic SVM.

We considered the weight of a false healthy diagnosis (patients that were diagnosed falsely as healthy by the classifier) to be more than that of a false unhealthy diagnosis. The former misses the chance to be diagnosed, which is the aim of this study, and the latter after referral to a DEXA scan or further investigation will be diagnosed as healthy. Thus, even though the accuracy and F1 score of SVM-Polynomial was lower than that of SVM-RBF, it may be a better classifier for our purpose.

Sollmann et al. showed a strong correlation between T2* and the BMD [19]. These researchers showed a significant positive correlation between R2(=1/T2), R2*(=1/T2*) and R2′(=R2* − R2) and the BMD in the coronal section of the femoral neck. The use of this parameter as a feature could potentially improve the classification accuracy.

The small influence of texture-based features in our study may be due to the different acquisition protocols used during MRI, which can affect the results of texture analysis [17]. Furthermore, the different measuring techniques used in different centers and by different manufacturers produce different patterns in the texture [17], which influences texture analysis. Thus, further investigation with a higher number of images with the same scan parameters and the same/similar machines is necessary to evaluate the importance of texture-based features in comparison to the geometric ones.

The SVM algorithm is suitable for small sample sizes. However, a small sample size, a large number of variables, and the inclusion of redundant variables, especially when samples cannot fully represent the distribution of the sample space, may lead to overfitting [17]. Borra and Di Siaccio [37,38] showed that 10-fold cross-validation and the parametric bootstrap estimator had the best performance in a simulation with varying signal-to-noise ratios and sample sizes (using regression trees, projection pursuit regression, and neural networks). The use of our validation technique (10-fold cross-validation) increases our confidence in the results.

Since it is still unclear which features are the most discriminative ones for the detection of osteoporosis from PFB images, we plan to employ deep neural network models using a higher number of images with the same scan parameters to identify the most appropriate features for this purpose in a future study. In this way, an end-to-end deep model can be utilized to perform both feature extraction and classification.

Despite its efficiency and effectiveness, this study has the following limitations.

First, when the number of databases grows, the current method will be unable to fulfill the needs of the medical community on a large scale because the use of engineering features based on machine learning techniques does not ensure the optimality of the feature vector in diverse scenarios. Second, if the database is expanded, deep learning techniques should be applied and analyzed. Third, in order to increase the amount of data, the performance of generative adversarial networks should be examined in future studies.

## 5. Conclusions

To our knowledge, this is the first study to carry out qualitative classification of human PFB using machine learning and genetic selection algorithms to identify subjects with unhealthy PFBs by reference to the results obtained from DEXA scans.

The SVM resulted in higher accuracy and was superior to the other algorithms. Geometric characteristics in femoral bone demonstrated the highest influence on patient discrimination. However, further investigation with a higher number of images of the same imaging protocol is suggested. Also, the effect of other segmentation techniques (such as deep convolutional neural networks) and other relevant features (such as T2*) on the classification performance should be investigated in future studies.

## Figures and Tables

**Figure 1 sensors-23-07612-f001:**
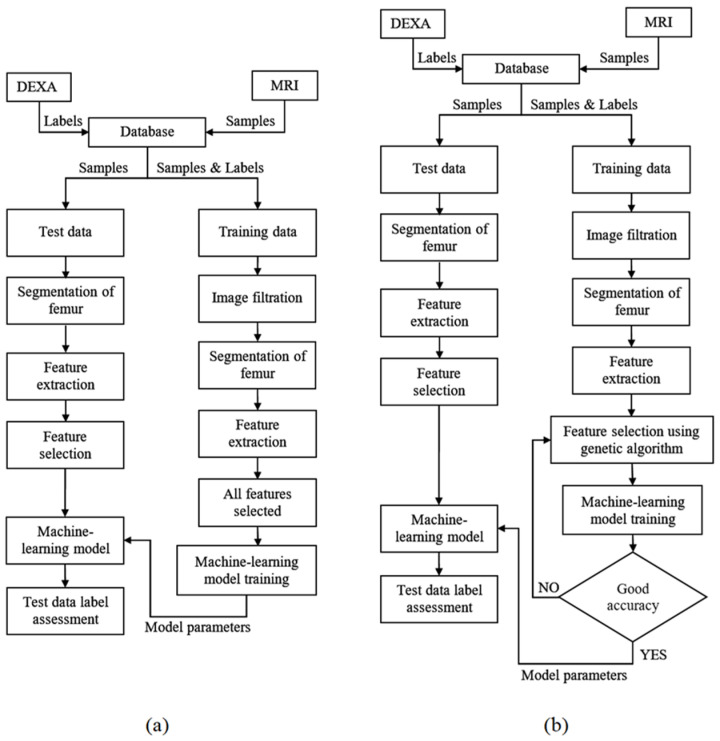
Overall procedure flowcharts. The proposed method for the diagnosis of patients at risk of osteoporosis using MRI and DEXA scans of the femur using different classifiers (**a**) and feature selection for classifiers using the Genetic algorithm (**b**).

**Figure 2 sensors-23-07612-f002:**
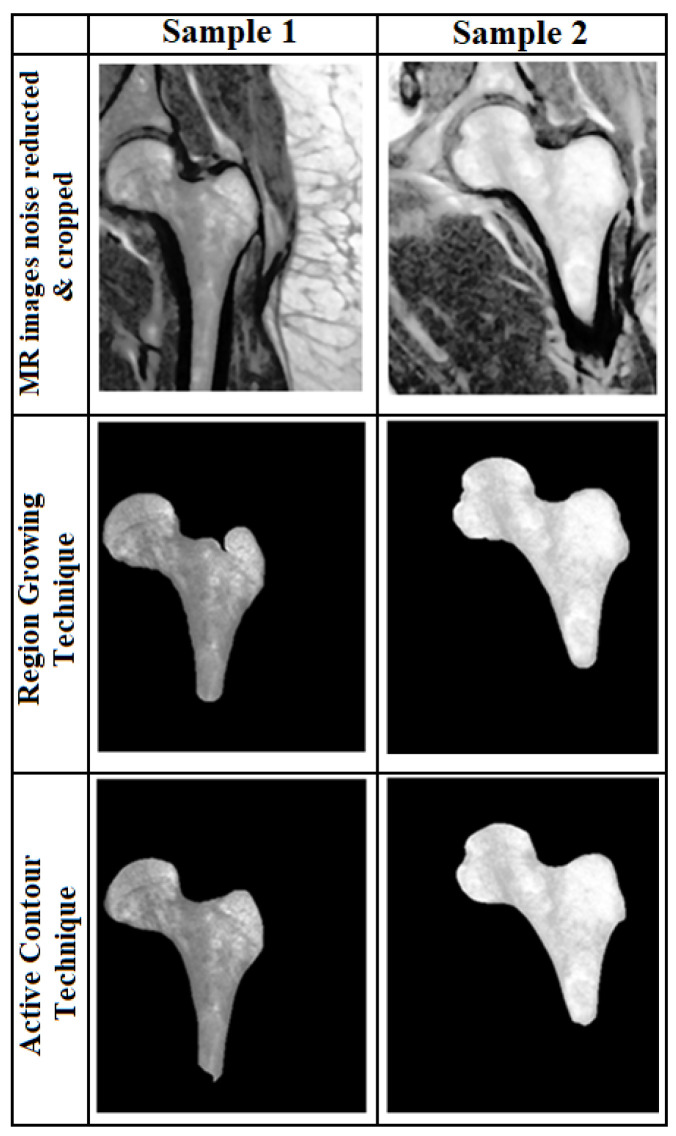
Segmentation results. Two segmentation results for two samples based on the active contour (**Bottom row**) and region growing (**middle row**) techniques.

**Figure 3 sensors-23-07612-f003:**
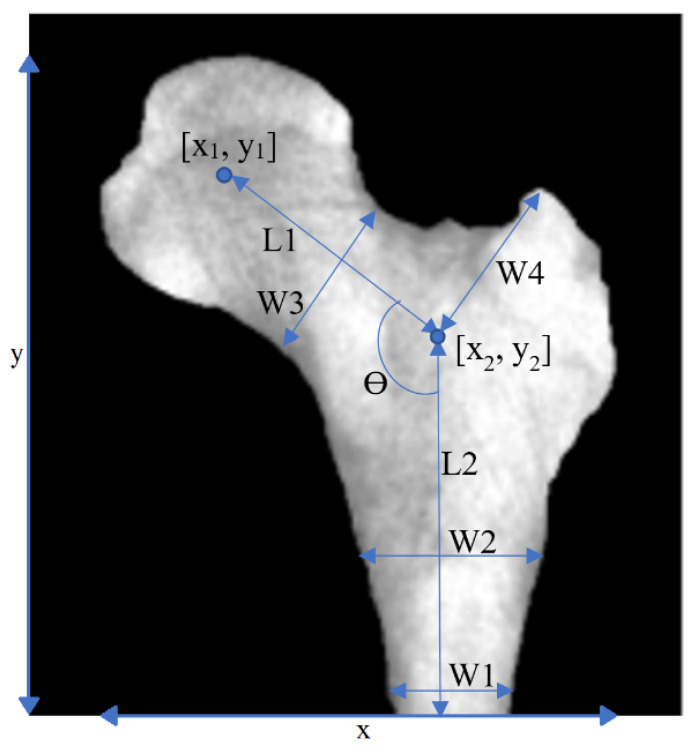
Geometrical features. Geometrical features extracted from a PFB image.

**Figure 4 sensors-23-07612-f004:**
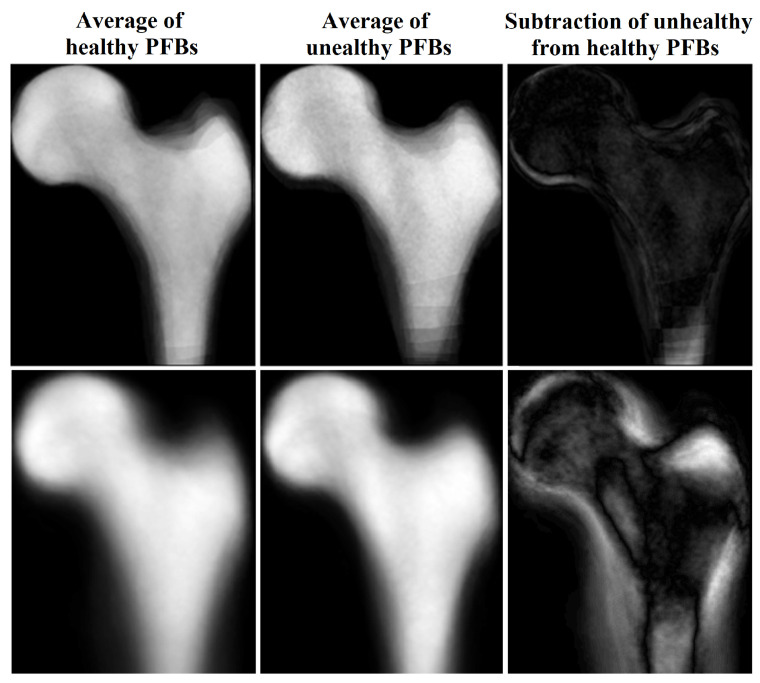
Average and subtraction of PFB images.

**Figure 5 sensors-23-07612-f005:**
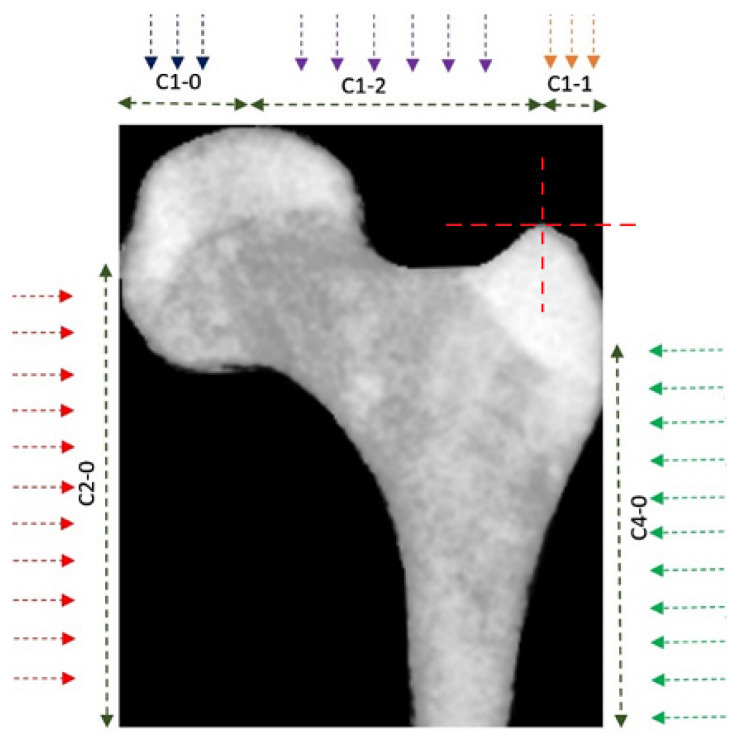
Feature curves for boundaries. Five feature curves were used to define the characteristics of the PFB’s boundaries.

**Figure 6 sensors-23-07612-f006:**
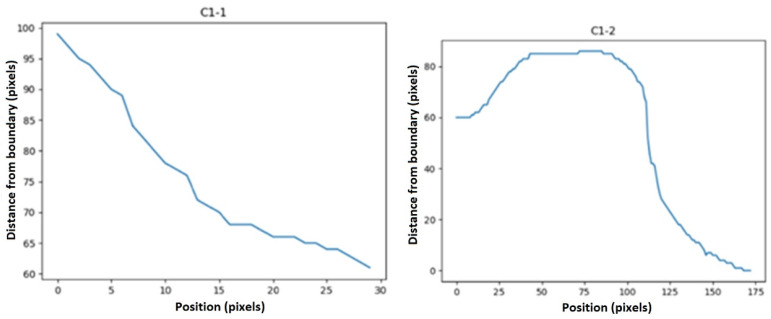
Sample of the resulting feature curves in Figure 5. Two samples of the resulting feature curves from the PFB in Figure 5.

**Figure 7 sensors-23-07612-f007:**
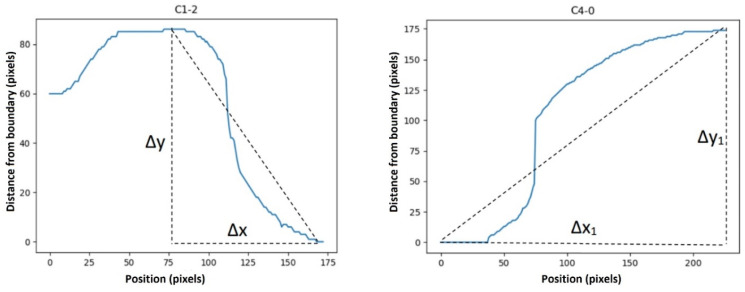
∆x, ∆y, ∆x_1, and ∆y_1 in Equations (3) and (4). ∆x, ∆y, ∆x_1, and ∆y_1 were defined as shown in the images above to calculate the “average ramp” and “first to last point ramp”.

**Figure 8 sensors-23-07612-f008:**
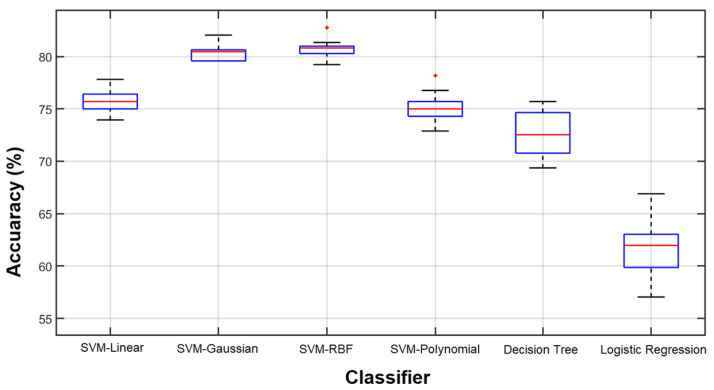
Classification results of 10 repetitions. Results obtained from 10 classification repetitions using various classifiers and all 204 features.

**Figure 9 sensors-23-07612-f009:**
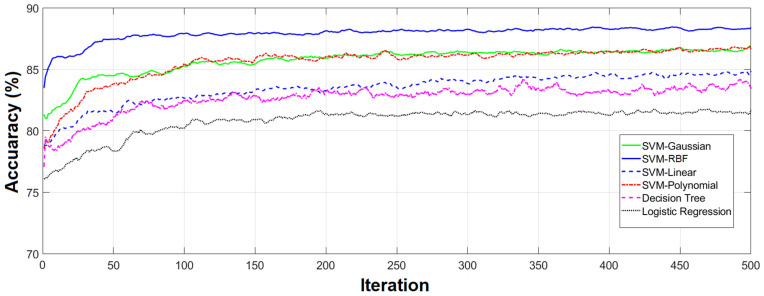
Classification results of 10 repetitions. Results obtained from repeating the classification 10 times using various classifiers and all 204 features.

**Figure 10 sensors-23-07612-f010:**
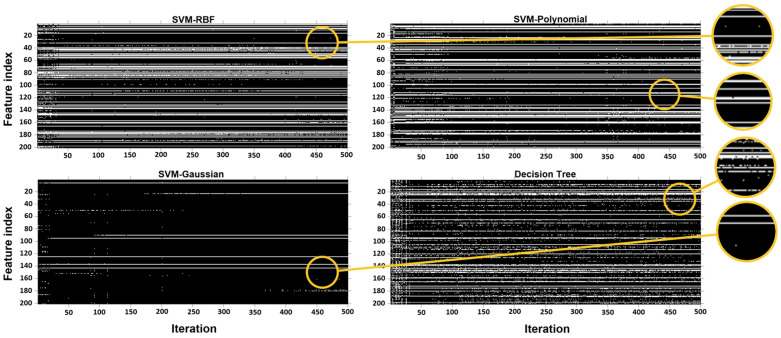
Representation of selected/missed features. Representation of selected/missed features during 500 iterations of the Genetic algorithm. White pixels represent a chosen feature. The feature index is based on the position of the features in Table 1, from left to right and up to down.

**Figure 11 sensors-23-07612-f011:**
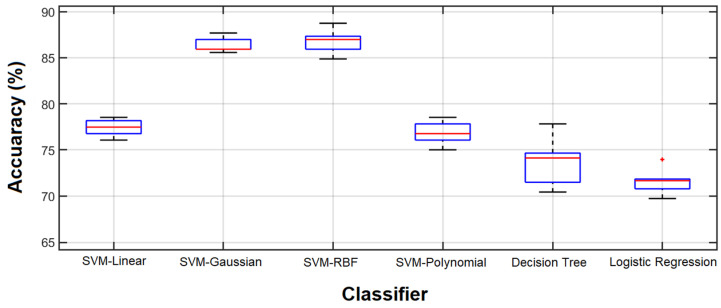
Classification accuracy. Classification accuracy using 56 features selected by the GA coupled with SVM-RBF.

**Figure 12 sensors-23-07612-f012:**
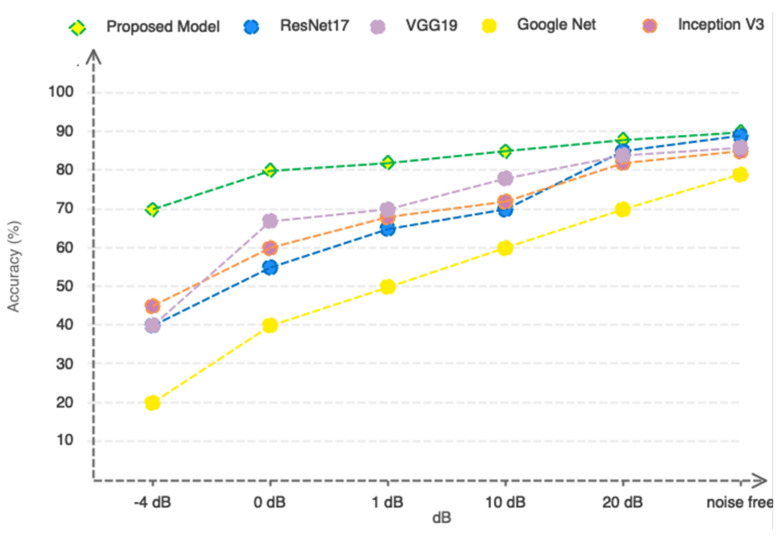
Comparing the performance of the proposed model with pre-trained networks in a noisy environment.

**Table 1 sensors-23-07612-t001:** Primary features. Primary extracted features used in this study (the second and third power of each feature was used as a feature as well).

Features	Description	No. of Features
**Geometrical features**	As in Figure 2, i.e.,: x1, y1, L1, L2, Ɵ, W0, W1, W2, W3,W4, x, y	**12**
**C1-1**	min, max, average ramp, kurtosis (ramp_removed_sig), skew (ramp_removed_sig), first to last point ramp, moment1 (ramped_remov_sig), moment 2 (ramp_removed_sig), variance, max. of second derivative	**10**
**C1-2**	min, max, mean, variance, kurtosis, skew, moment1, moment2	**8**
**C1-0**	min, max, average ramp, kurtosis (ramp_removed_sig), skew (ramp_removed_sig), first to last point ramp, moment1 (ramped_remov_sig), moment 2 (ramp_removed_sig), variance, max of second derivative	**10**
**C2-0**	min, max, average ramp, kurtosis (ramp_removed_sig), skew (ramp_removed_sig), first to last point ramp, moment 1 (ramped_remov_sig), moment2 (ramp_removed_sig), variance, max of second derivative	**10**
**C4-0**	min, max, average ramp, kurtosis (ramp_removed_sig), skew (ramp_removed_sig), first to last point ramp, moment1 (ramped_remov_sig), moment2 (ramp_removed_sig), variance, max of second derivative	**10**
**Features based on texture**	mean, variance, skew	**3**
**Fractal dimensions**	fractal dimensions	**1**
**Features based on texture**	gray-level co-occurrence matrix (GLCM)	**4**

**Table 2 sensors-23-07612-t002:** Final classification results. Results based on each model using a confusion matrix (TH, True Healthy; FH, False Healthy; TU, True Unhealthy; FU, False Unhealthy; Acc., Accuracy (TH + TU)/(TH + TU + FH + FU)).

Features	Classifier	TH	FH	TU	FU	Acc. (%)	F1 Score (%)	No. of Selected Features
**Using selected features by the Genetic algorithm**	SVM-Linear	167	24	75	18	85.21	83.48	60
	**SVM-RBF**	**172**	**18**	**81**	**13**	**89.08**	**87.84**	**56**
	**SVM-Gaussian**	**173**	**24**	**75**	**12**	**87.32**	**85.61**	**7**
	**SVM-Polynomial**	**161**	**11**	**88**	**24**	**87.68**	**86.80**	**54**
	Decision Tree	168	24	75	17	85.56	83.83	62
	Logistic Regression	158	24	75	27	82.04	80.37	80

**Table 3 sensors-23-07612-t003:** Comparing the effectiveness of the proposed model with other networks.

Methods	ACC. (%)	F1 Score (%)	Time per Iteration
ResNet17	89.05	87.14	4.2 s
VGG19	88.90	89.1	7.1 s
GoogleNet	79.47	80.36	9 s
Inception v3	85.04	86.74	8.5 s
**Proposed Model**	**89.08**	**87.84**	1 s

## Data Availability

We have made our dataset available to the public at https://github.com (file name: Mojtab2023/Classification-of-Proximal-Femoral-Bone-Using-Geometric-Features-and-Texture-Analysis-in-MR-Images-f (1 January 2022).

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
