# Peer review of "Qualitative Classification of Proximal Femoral Bone Using Geometric Features and Texture Analysis in Collected MRI Images for Bone Density Evaluation"

_sensors, 2023, doi:10.3390/s23177612_

Round 1
Reviewer 1 Report
This work proposes a qualitative classification method of proximal femoral bone for bone density evaluation using MRI. Some issues undermine the methodological reporting of the paper and should be addressed by the authors:
-Introduction: according to the authors, this method could be helpful to detect patients referred to imaging centers “for other complications”. The sentence should be specified.
- Methods: The number of images and patients are a result of your inclusion and exclusion criteria. Please, specify how you have enrolled your patients, and whether some patients were excluded (and why).
Provide a flowchart of the study.
Once the patients were selected, for each how and who chose the images used in the study?
How much time passed between MRI and DEXA? Was it predetermined? Report the average with the range.
Who evaluated the correctness of the segmentations? Was it the same person who performed the manual segmentations? What expertise does he or she have?
- Discussion: Be consistent with the abbreviations (Support vector machine was abbreviated in SVM).
Dedicate a part of the discussion to the limitations of this study. To facilitate that, carry out a self-assessment using the RQS tool developed by Lambin et al. (https://doi.org/10.1038/nrclinonc.2017.141) and then compare your result with the mean reported in literature (https://doi.org/10.1007/s00330-022-09187-3). It would be useful to highlight pro and cons of your methodology.
Minor:
Please proofread the paper for typos such as on Page 3 “divdied” “pixles” and so on.
The paper is well-written. Only a few typos need a revision.
Author Response
Reviewer#1:
Comments:
This work proposes a qualitative classification method of proximal femoral bone for bone density evaluation using MRI. Some issues undermine the methodological reporting of the paper and should be addressed by the authors:
- ⎫ Thanks to the esteemed reviewer, we believe that your comments have been very useful and effective in enhancing the scientific and writing framework of the manuscript. We have considered all the comments in their entirety and made every effort to correct the manuscript in the manner suggested by the honorable reviewer.
- 1. Introduction: according to the authors, this method could be helpful to detect patients referred to imaging centers “for other complications”. The sentence should be specified.
- ⎫ The manuscript is revised based on this comment. Yes, the opinion of the honorable reviewer is absolutely correct. We have modified the relevant sentence as follows:
“If we can determine bone quality from MRI, when patients are referred to imaging centers to check the health of body organs such as bladder, intestine, liver, kidneys, ovaries and breast in women and prostate in men, we can detect many patients with osteoporosis or on the verge of developing osteoporosis earlier and use appropriate treatment to reduce the number of patients in the society.”
Which is highlighted in Introduction section, page 2 and lines 67-71.
- 2. The number of images and patients are a result of your inclusion and exclusion criteria. Please, specify how you have enrolled your patients, and whether some patients were excluded (and why).
- ⎫ The manuscript is revised based on this comment. The point to remember when recording MRI scans is that we did not take into account the number of images obtained based on the number of participants invited to the test. Patients with various diseases were referred to the imaging center by their doctors, and the purpose of the test and the expected results were explained to the patients during the patient admission process. We used the MRI scans taken from them in our research, and as an incentive gift due to the subjects' consent, a free DEXA image was obtained from each patient after taking the MRI scans, as well as information about the participants' BMD and bone condition. The necessary information was provided to the participants for free in the experiment based on the DEXA images obtained.
- ⎫ People who went to the imaging center with a doctor's referral and did not consent to use their images in this research after explaining the project, were left out of the testing process.
- ⎫ In general, these patients are excluded from the testing process: lack of personal consent for testing, people who used a pacemaker, people who had an implant or a metal foreign body in their body (such as having a metal piece in the brain, eye or platinum in bone).
Which is highlighted in section 2-1, page 2 and lines 95-127.
- 3. Provide a flowchart of the study.
- ⎫ Due to the According to the reviewer's opinion, the flowchart of the proposed method is shown in Figure 1.
Figure 1. Overall procedure Flowcharts; the proposed method for the diagnosis of patients at risk of osteoporosis using MRI and DEXA scans of the femur using different classifiers (a) and feature selection for classifiers using genetic algorithm (b).
- 4. Once the patients were selected, for each how and who chose the images used in the study?
- ⎫ The manuscript is revised based on this comment. If the image obtained after viewing by the center's technician is of sufficient quality cand does not contain unacceptable noise or artifacts, it is chosen for review and analysis.
Which is highlighted in section 2-1, page 2 and lines 95-127.
- 5. How much time passed between MRI and DEXA? Was it predetermined? Report the average with the range.
- ⎫ The manuscript is revised based on this comment. The time it took to obtain MRI images of the femur was approximately 15 minutes, and the time it took to obtain the BMD of the femur using the DEXA technique was approximately 2 minutes.
Which is highlighted in section 2-1, page 2 and lines 95-127.
- 6. Who evaluated the correctness of the segmentations? Was it the same person who performed the manual segmentations? What expertise does he or she have?
- ⎫ The manuscript is revised based on this comment. First, a doctor with expertise in radiology performs manual segmentation. The accuracy of our segmentation is then validated by the radiologist's segmentation.
- ⎫ The person who recorded the MRI images had an MRI fellowship (specialized in radiology) and the person who recorded the DEXA was a radiologist.
Which is highlighted in section 2-1, page 2 and lines 95-127.
- 7. Discussion: Be consistent with the abbreviations (Support vector machine was abbreviated in SVM).
- ⎫ The manuscript is revised based on this comment. According to the reviewer's opinion, the acronym SVM was placed in the manuscript instead of the word support vector machine.
- 8. Dedicate a part of the discussion to the limitations of this study. To facilitate that, carry out a self-assessment using the RQS tool developed by Lambin et al. (https://doi.org/10.1038/nrclinonc.2017.141) and then compare your result with the mean reported in literature (https://doi.org/10.1007/s00330-022-09187-3). It would be useful to highlight pro and cons of your methodology.
- ⎫ The manuscript is revised based on this comment. Yes, the opinion of the honorable reviewer is absolutely correct. However, due to time constraints, we have tried to express the limitations of this research in a textual way.
- ⎫ Despite its efficiency and effectiveness, this study has the following limitations:
First, when the number of databases grows, the current method will be unable to fulfill the needs of the medical community on a large scale, because the use of engineering features based on machine learning techniques does not ensure the optimality of the feature vector in diverse scenarios. Second, if the database is expanded, deep learning techniques should be applied and analyzed. Third, in order to increase data, the performance of generative adversarial network should be examined in future studies.
Which is highlighted in Discussion section, page 9 and lines 422-428.
- 9. Please proofread the paper for typos such as on Page 3 “divdied” “pixles” and so on.
- ⎫ The manuscript is revised based on this comment. According to the reviewer's opinion, we corrected the spelling mistakes in the manuscript.

Reviewer 2 Report
The submitted manuscript compared the performance of several machine learning methods to discriminate healthy and unhealthy femurs using collected MRI images, and genetic algorithm was used to identify the importance feature. It doesn’t surprise me that SVM with Gaussian and RBF kernel perform better than the other machine learning methods. Below are some comments that may be helpful to improve the quality of the manuscript.
1. The number of patients labeled as healthy (185) is larger than these labeled as unhealthy (99). In other word, the data used in this study is mild imbalanced. How did you address this issue? Could you also specify the number of healthy and unhealthy patients for each scan parameters?
2. In section 2.4 Segmentation, two methods were used for segmentation and the better result was selected (line 129-130). Could you clarify the metric you used to evaluate the quality of segmentation?
3. Fig. 3 needs to be described in details. Please clarify the meaning of each geometrical feature, and the methods used to extract these features.
4. In Fig. 5 and 6, the unit of the five descriptive curves is pixel. Pixel itself doesn’t provide meaningful information since different images usually have different spatial resolution. I think the authors should convert the unit from pixel to mm (millimeter) for further processing.
5. In Fig. 4, it is as expected that the difference of unhealthy from healthy averaged PFBs with same scan parameters is mainly in the boundaries. I think this is mainly coming from the misalignment during the average. For the same scan parameters, if the authors randomly select half of the PFBs, average them and compare the difference the averaged image with the other half, I expect the difference is mainly in the boundaries, too.
6. The geometric features, as well as the five descriptive curves highly depend on the individual conditions of each patient, such as the weight, height. Why are these features related to the bone density?
7. Could you provide more detials about the comparssion with the deep-learing based method ?
Author Response
Reviewer#2:
Comments:
The submitted manuscript compared the performance of several machine learning methods to discriminate healthy and unhealthy femurs using collected MRI images, and genetic algorithm was used to identify the importance feature. It doesn’t surprise me that SVM with Gaussian and RBF kernel perform better than the other machine learning methods. Below are some comments that may be helpful to improve the quality of the manuscript.
- ⎫ Thanks to the esteemed reviewer, we believe that your comments have been very useful and effective in enhancing the scientific and writing framework of the manuscript. We have considered all the comments in their entirety and made every effort to correct the manuscript in the manner suggested by the honorable reviewer.
- 1. The number of patients labeled as healthy (185) is larger than these labeled as unhealthy (99). In other word, the data used in this study is mild imbalanced. How did you address this issue? Could you also specify the number of healthy and unhealthy patients for each scan parameters?
- ⎫ The manuscript is revised based on this comment. Yes, the opinion of the respected reviewer is absolutely correct, in order to solve this problem, we artificially increased the data of the minority class using data augmentation technique. In this way, the rotation range of the images was considered to be 40 degrees. Also, the horizontal shift of the images, the vertical shift of the images is considered to be 0.2%.
- ⎫ The manuscript is revised based on this comment. With respect to the opinion of the respected referee, considering that the scan parameters were collected from 3 different centers, for this reason, they are as follows for sick and non-sick people:
•T1/TR=415.00ms/TE=19.00ms/Flip-Angle=150 (44 patients, 33 healthy, 11 unhealthy)
•T1/TR=536.00ms/TE=11.00ms/Flip-Angle=180 (43 patients, 31 healthy, 12 unhealthy)
•T1/TR=4070.00ms/TE=33.00ms/Flip-Angle=180 (42 patients, 29 healthy, 13 unhealthy)
•T1/TR=420.00ms/TE=22.00ms/Flip-Angle=180 (45 patients, 36 healthy, 9 unhealthy)
•T2/TR=3600.00ms/TE=80.00ms/Flip-Angle=150 (45 patients, 19 healthy, 26 unhealthy)
•T2/TR=7840.00ms/TE=109.00ms/Flip-Angle=150 (41 patients, 28 healthy, 13 unhealthy)
•T1/TR=389.00ms/TE=13.42ms/Flip-Angle=110 (24 patients, 9 healthy, 15 unhealthy)
Which is highlighted in section 2-1, page 2 and lines 95-127.
- 2. In section 2.4 Segmentation, two methods were used for segmentation and the better result was selected (line 129-130). Could you clarify the metric you used to evaluate the quality of segmentation?
- ⎫ The manuscript is revised based on this comment. The index used to evaluate 2 different segmentation methods is Dice Similarity Coefficient (DSC) and Jaccard index (also called Intersection over Union, IoU). These two indicators can be defined as follows:
The segmentation outcomes were numerically evaluated using the DSC and IoU. The measurements are specified according to (1) and (2).
|
(1) |
|
|
(2) |
|
Where S and T, correspondingly, represent the segmentation result and the ground truth [1, 2].
[1] Chay, Z. E.; Lee, C. H.; Lee, K. C.; Oon, J. S.; Ling, M. H., Russel and Rao coefficient is a suitable substitute for Dice coeffi-cient in studying restriction mapped genetic distances of Escherichia coli. arXiv preprint arXiv:2302.12714 2023.
[2] Qingyun, F.; Zhaokui, W., Fusion Detection via Distance-Decay Intersection over Union and Weighted Dempster–Shafer Evidence Theory. Journal of Aerospace Information Systems 2023, 20 (3), 114-125.
Which is highlighted in section 2-4, page 4 and lines 161-170.
- 3. Fig. 3 needs to be described in details. Please clarify the meaning of each geometrical feature, and the methods used to extract these features.
- ⎫ The manuscript is revised based on this comment. In Figure 3, the parameters W1, W2, and so on are defined as geometric features. The goal of this definition of the geometric parameters marked on the figure was to see if there was a link between the defined geometric parameters and osteoporosis. To put it another way, we define parameters W1, W2, and so on as geometric features rather than as abbreviations for specific phrases or words.
Which is highlighted in section 2-4, page 5 and lines 171-177.
- 4. In Fig. 5 and 6, the unit of the five descriptive curves is pixel. Pixel itself doesn’t provide meaningful information since different images usually have different spatial resolution. I think the authors should convert the unit from pixel to mm (millimeter) for further processing.
- ⎫ The manuscript is revised based on this comment. According to the reviewer's opinion, Figure 6 was changed from pixels to millimeters.
- 5. In Fig. 4, it is as expected that the difference of unhealthy from healthy averaged PFBs with same scan parameters is mainly in the boundaries. I think this is mainly coming from the misalignment during the average. For the same scan parameters, if the authors randomly select half of the PFBs, average them and compare the difference the averaged image with the other half, I expect the difference is mainly in the boundaries, too.
- ⎫ With respect to the opinion of the honorable reviewer, the non-alignment of the average has nothing to do with this issue. We created an averaged image from unhealthy (based on DEXA results) PFBs and another averaged image from healthy PFBs in our training dataset; after flipping (for right femurs), cropping and framing PFB images at similar positions. For images with same scan parameters, the difference between these two images indicated the important regions in which these two classes differ the most, i.e. the boundaries of the PFB. This means that characteristics of the boundaries may represent the difference between the two PFB classes. When we used all images with different scan parameters, the inside region also became more apparent in the subtracted image.
- 6. The geometric features, as well as the five descriptive curves highly depend on the individual conditions of each patient, such as the weight, height. Why are these features related to the bone density?
- ⎫ Based on previous studies [1], this study examines the relationship between physical shape and bone problems. Because the possibility of fracture is related to osteoporosis. Therefore, it was decided to investigate the possibility of a relationship between bone physics and osteoporosis, and the results confirmed the existence of a relationship between bone physics and osteoporosis.
[1] El-Kaissi S, Pasco JA, Henry MJ, Panahi S, Nicholson JG, Nicholson GC, Kotowicz MA. Femoral neck geometry and hip fracture risk: the Geelong osteoporosis study. Osteoporosis international. 2005 Oct;16:1299-303.
- 7. Could you provide more detials about the comparssion with the deep-learing based method?
- ⎫ The manuscript is revised based on this comment. Based on the reviewer's opinion, we provided more results to compare the proposed model based on engineering feature selection/extraction with deep learning networks. For this purpose, the following content is provided:
“Due to the nature of MRI imaging, the slightest movement of the patient can cause noise in the captured images. Accordingly, the proposed method should be designed in such a way that it can be resistant to movement and environmental noises. In order to evaluate the proposed model more accurately, we have artificially added Gaussian white noise in a wide range of different SNRs to the images and presented the results based on the compared ResNet17, VGG19, GoogleNet, and Inception v3 pre-trained deep networks in Figure 12. According to Figure 12, as can be seen, the proposed model based on the selection/extraction of engineering features can be more resistant to noise than ResNet17, VGG19, GoogleNet, and Inception v3 networks in a wide range of different SNRs. So, in SNR=-4, it still remains above 80%.
Figure 12. Comparing the performance of the proposed model with pre-training networks in noisy environment.”
Which is highlighted in results, page 7 and lines 355-364.

Reviewer 3 Report
I have received the manuscript entitled " Qualitative Classification of Proximal Femoral Bone Using Geometric Features and Texture Analysis in Collected MRI Images for Bone Density Evaluation" for review and have found it to be very interesting and very well-written.
I made very minor linguistic corrections, and some specific data presentation suggestions. All of these are highlighted "in YELLOW colour" inside the pdf file. By double-clicking on any highlighted text, the authors will find “inside a balloon window” a correction or a suggestion or a concern for clarification.
I believe that the authors have interesting general aim, scope and results included in the manuscript. Therefore, the current state of the manuscripts is suitable for publication.
Perhaps, my major comment was on the introduction paragraph that may need an expansion by citing more works and references that focus on the importance of the study. Some other comments were aimed to produce better presentation of text paragraphs and some figures. The latter are all highlighted inside the pdf file.
Apart from that, the literature review is adequate but needs a little bit of expansion, the aims and methods are well explained, the contributions are well-stated, the mathematical models are adequately explained, the parameters used in the analysis may require a little bit of explanation and clarification, the GA selection criteria or approach may need more clarification, the experimental and simulation setups are clear, the results are well-presented and the thoroughness of the conclusion section is adequate.
Exact comments/queries/suggestions to enhance the work are highlighted inside the pdf file.

Very minor linguistic enhancements are suggested inside the pdf file.
Author Response
Reviewer#3:
Comments:
I have received the manuscript entitled "Qualitative Classification of Proximal Femoral Bone Using Geometric Features and Texture Analysis in Collected MRI Images for Bone Density Evaluation" for review and have found it to be very interesting and very well-written.
I made very minor linguistic corrections, and some specific data presentation suggestions. All of these are highlighted "in YELLOW colour" inside the pdf file. By double-clicking on any highlighted text, the authors will find “inside a balloon window” a correction or a suggestion or a concern for clarification.
I believe that the authors have interesting general aim, scope and results included in the manuscript. Therefore, the current state of the manuscripts is suitable for publication.
- ⎫ Thanks to the esteemed reviewer, we believe that your comments have been very useful and effective in enhancing the scientific and writing framework of the manuscript. We have considered all the comments in their entirety and made every effort to correct the manuscript in the manner suggested by the honorable reviewer.
- ⎫ We have applied all the highlights you specified in the PDF to the manuscript. Some of your comments in the highlighted PDF needed an explanation, and here we have provided your comment along with the answer. Thank you very much for taking the time to review our manuscript.
- 1. Perhaps, my major comment was on the introduction paragraph that may need an expansion by citing more works and references that focus on the importance of the study. Some other comments were aimed to produce better presentation of text paragraphs and some figures. The latter are all highlighted inside the pdf file.
- ⎫ The manuscript is revised based on this comment. Yes, the opinion of the honorable reviewer is absolutely correct. According to the reviewer's opinion, we have reviewed more studies on different types of osteoporosis, current diagnosis guidelines and diagnosis problems in this section. Added content is highlighted.
- ⎫ “Osteoporosis can be considered as a result of lack of calcium as well as other minerals in the bones. This disease can be classified into the following 3 categories: a) primary osteoporosis including idiopathic osteoporosis, type 1 osteoporosis due to estrogen deficiency common in postmenopausal women, type 2 osteoporosis due to estrogen deficiency with loss Bone density. Vertebral fractures can be considered as one of the common consequences of osteoporosis, which leads to constant pain, skeletal-muscular abnormalities, respiratory and deterioration of physical performance and quality of life. Also, there will be a chance of sustaining a fracture after an initial fracture. Physiotherapy can be considered a special non-pharmacological method, which can be useful and effective for patients who have osteoporosis, through manual methods, resistance exercises, etc.”
- ⎫ “Bone Mineral Density (BMD) is currently the main factor for determining bone quality and risk of osteoporosis. Dual Energy X-ray absorptiometry (DEXA) is considered the clinical gold standard for BMD evaluation. According to the guidelines of the National Osteoporosis Foundation, the indications for DEXA examination are: a) Women 65 years old and men 70 years and older. b) Postmenopausal women and men over 50 who have suffered bone fractures. c) Adults who suffer from rheumatoid arthritis or use medications such as glucocorticoids and have low bone mass.”
Which is highlighted introduction, pages 1-2 and lines 31-41 and 48-54.
- 2. I tried to search and locate the file on the github website, but did not find it... if it is necessary to keep this piece of information in the paper, please included the URL of the file, or alternatively, supply the file to the publisher as "supplementary material".
- ⎫ Yes, the opinion of the honorable reviewer is absolutely correct. We are waiting for the manuscript to be printed so that we can make the link available after it is printed. Making the link available before printing can cause an infringement of interests.
- 3. In this sub-section 2.4, try to place text paragraphs as close as possible to figures that they refer to. I found it hard to follow text descriptions while scrolling up and down to figures.
- ⎫ According to the reviewer's opinion, we have made the figures closer to the explanations for more accurate and easy reading.
- 4. It will be great to state the medical or expert terminologies of the parameters marked in this figure... for instance what is the term for W2, W4, L1, ... etc.
- ⎫ The manuscript is revised based on this comment. In Figure 3, the parameters W1, W2, and so on are defined as geometric features. The goal of this definition of the geometric parameters marked on the figure was to see if there was a link between the defined geometric parameters and osteoporosis. To put it another way, we define parameters W1, W2, and so on as geometric features rather than as abbreviations for specific phrases or words.
Which is highlighted in section 2-4, page 5 and lines 171-177.
- 5. Which factors in your study were independent? Please add a table to state and clarify this independence.
- ⎫ With respect to the opinion of the respected reviewer, we have already introduced the extracted features (variables) in Table 1.
- 6. I suggest stating what is T2* parameters! ** The same suggestion for the other parameters below (R2, R2', R2*).
- ⎫ With respect to the opinion of the respected reviewer, T1, T2, etc. are scan parameters that are explained in the data collection section.
- 7. Apart from that, the literature review is adequate but needs a little bit of expansion, the aims and methods are well explained, the contributions are well-stated, the mathematical models are adequately explained, the parameters used in the analysis may require a little bit of explanation and clarification, the GA selection criteria or approach may need more clarification, the experimental and simulation setups are clear, the results are well-presented and the thoroughness of the conclusion section is adequate.
- ⎫ The manuscript is revised based on this comment. With respect to the reviewer's opinion, the requested items have already been given in the manuscript. For example, the initial settings in the GA algorithm are considered as follows:
In the GA, the total number of population was set to 100, with a maximum of 500 iterations. The flowchart of this procedure is shown in Figure 1(b). In our study, we used a cost function that would result in the best accuracy with the least number of features (NOF) as defined below:
|
(6) |
|
- ⎫ However, with respect to the opinion of the respected reviewer, the feature selection method is discussed in more detail as follows in the manuscript:
Table 2 shows the best results (confusion matrix) achieved based on the features selected by the GA for each classifier and its corresponding number of selected features. Ac-cording to Table 2, as can be seen, the GA algorithm has selected different features and the number of features for different categories. For example, 60 features, 56 features, and 7 features are selected for SVM-Linear, SVM-RBF, and SVM-Gaussian, respectively. The types of features selected by GA are discussed below.
Table 2. Final classification results; Results based on each model using confusion matrix (TH: True Healthy, FH: False Healthy, TU: True Unhealthy, FU: False Unhealthy and Acc.: Accuracy ((TH + TU) / (TH + TU + FH + FU)).
|
Features |
Classifier |
TH |
FH |
TU |
FU |
Acc. (%) |
F1 score (%) |
No. of selected features |
|
Using selected features by genetic algorithm |
SVM-Linear |
167 |
24 |
75 |
18 |
85.21 |
83.48 |
60 |
|
SVM-RBF |
172 |
18 |
81 |
13 |
89.08 |
87.84 |
56 |
|
|
SVM-Gaussian |
173 |
24 |
75 |
12 |
87.32 |
85.61 |
7 |
|
|
|
SVM-Polynomial |
161 |
11 |
88 |
24 |
87.68 |
86.80 |
54 |
|
Decision Tree |
168 |
24 |
75 |
17 |
85.56 |
83.83 |
62 |
|
|
Logistic Regression |
158 |
24 |
75 |
27 |
82.04 |
80.37 |
80 |
In the SVM-Gaussian, the 7 selected features were W0, C1-1 (max. of second derivative), C1-1(variance) 2, C2-0 (first to last point ramp) 2, C4-0 (moment2 (ramp_removed_sig)) 2 and fractal dimensions. On the other hand, main selected features for SVM-RBF were based on geometric features (15 features) and C2-0 (9 features). Most influential features in our study by a descending order were geometrical features (in table 1), C1-2, C2-0, fractal dimensions, C1-0, C1-1 and finally features based on the texture. Most used feature across all classifiers were y1, W2 and C1-1 (kurtosis (ramp_removed_sig)).
Although every primary feature was used either in its first, second or third order, a total of 105 features were not used by any classifier; half of which belonged to the 3rd power of the primary features. This was expected as we designed the cost function of the GA not only to improve the classification performance, but also to reduce the dimensionality of feature vector.
Each run of the GA may result in a different set of features. Figure 10 shows the selection process of the GA for SVM-RBF, SVM-Polynomial, SVM-Gaussian and decision tree during the 500 iterations. Each of these binary images have 500 columns (corresponding to 500 iterations) and 204 rows (associated with 204 features). Ones (white pixels) and zeros (black pixels) represent selected or missed corresponding feature in the relevant iteration. The horizontal white line represents a feature that was steadily chosen by the GA. The most unstable accuracy based on feature selection by the GA was for the decision tree model with less than 2 % fluctuations in accuracy.
- 8. Exact comments/queries/suggestions to enhance the work are highlighted inside the pdf file.
- ⎫ The manuscript is revised based on this comment. According to the reviewer's opinion, we have corrected all the highlighted items in the PDF and changed them based on the reviewer's opinion. All requested changes are highlighted in the manuscript.

Reviewer 4 Report
Dear Authors,
Given the high prevalence and morbidity rates of osteoporosis in worldwide aging populations, I think that this manuscript depicts some intriguing findings. However, I am writing to express my concerns. Indeed, while the study addresses an important topic in the field of osteoporosis diagnosis and presents potential advancements in bone quality evaluation, there are several issues that give rise to critical concerns, which are provided below.
Major concerns
INTRODUCTION: the section provides a comprehensive overview of osteoporosis and the need for early diagnosis. However, it would greatly benefit from references to current evidence-based guidelines and established diagnostic modalities such as Dual Energy X-ray Absorptiometry (DEXA) scans for assessing bone mineral density (BMD). Moreover, it might be interesting to discuss the implication of osteoporosis in physical functioning and quality of life in order to underline the need for a precise and sustainable assessment. In accordance, you should cite the following references:
- Holubiac IȘ, Leuciuc FV, Crăciun DM, Dobrescu T. Effect of Strength Training Protocol on Bone Mineral Density for Postmenopausal Women with Osteopenia/Osteoporosis Assessed by Dual-Energy X-ray Absorptiometry (DEXA). Sensors (Basel). 2022;22(5):1904. Published 2022 Feb 28. doi:10.3390/s22051904
- de Sire A, Lippi L, Venetis K, et al. Efficacy of Antiresorptive Drugs on Bone Mineral Density in Post-Menopausal Women With Early Breast Cancer Receiving Adjuvant Aromatase Inhibitors: A Systematic Review of Randomized Controlled Trials. Front Oncol. 2022;11:829875. Published 2022 Jan 21. doi:10.3389/fonc.2021.829875
- Yang L, Chen C, Zhang Z, Wei X. Diagnosis of Bone Mineral Density Based on Backscattering Resonance Phenomenon Using Coregistered Functional Laser Photoacoustic and Ultrasonic Probes. Sensors (Basel). 2021 Dec 9;21(24):8243. doi: 10.3390/s21248243.
METHODS: in my opinion, the whole section should be extensively reviewed. Indeed, reviewing the section rises several critical concerns, as it follows:
- The sample size calculation should be better characterized, providing the rationale behind selecting the specific number of subjects and imaging centers. It is essential to provide a clear justification for the chosen sample size, considering the statistical power and representativeness of the study population.
- The 3-year period of investigation should be better clarified.
- The study lacks critical information about the inclusion and exclusion criteria for the population assessed. Indeed, this is a crucial element which could help preventing sources of bias (e.g., selection bias).
- The demographic characteristics of the participants should be implemented. You might consider including further information (e.g., age, sex, BMI, comorbidities, etc.) within an additional table.
DISCUSSION: while the results demonstrate promising accuracy rates, it is important to discuss the limitations of the study. Keeping in mind the aforementioned suggestions, I think that the section should be implemented addressing potential sources of bias, such as variations in MRI scan parameters. This would enhance the generalizability of your proposed algorithm. Moreover, further discussion on the clinical implications and potential applications of your proposed algorithm should be provided. Indeed, considering the ongoing advancements in the field, it would be beneficial to discuss your findings in the context of existing literature.
ETHICS STATEMENT: please, provide the manuscript’s IRB.
Minor concerns
WHOLE MANUSCRIPT: please, note that numbers at the beginning of a sentence should be spelled (e.g., lines 14, 17, 113, etc.). Otherwise, you might consider starting the sentence with a connector.
INTRODUCTION: lines 57-59. Apparently, the sentence should be changed as it follows: “[…] was practical or not”.
WHOLE MANUSCRIPT: please, note that numbers at the beginning of a sentence should be spelled (e.g., lines 14, 17, 113, etc.). Otherwise, you might consider starting the sentence with a connector.
INTRODUCTION: lines 57-59. Apparently, the sentence should be changed as it follows: “[…] was practical or not”.
Author Response
Reviewer#4:
Comments:
Given the high prevalence and morbidity rates of osteoporosis in worldwide aging populations, I think that this manuscript depicts some intriguing findings. However, I am writing to express my concerns. Indeed, while the study addresses an important topic in the field of osteoporosis diagnosis and presents potential advancements in bone quality evaluation, there are several issues that give rise to critical concerns, which are provided below.
- ⎫ Thanks to the esteemed reviewer, we believe that your comments have been very useful and effective in enhancing the scientific and writing framework of the manuscript. We have considered all the comments in their entirety and made every effort to correct the manuscript in the manner suggested by the honorable reviewer.
- 1. INTRODUCTION: the section provides a comprehensive overview of osteoporosis and the need for early diagnosis. However, it would greatly benefit from references to current evidence-based guidelines and established diagnostic modalities such as Dual Energy X-ray Absorptiometry (DEXA) scans for assessing bone mineral density (BMD). Moreover, it might be interesting to discuss the implication of osteoporosis in physical functioning and quality of life in order to underline the need for a precise and sustainable assessment. In accordance, you should cite the following references:
- Holubiac IȘ, Leuciuc FV, Crăciun DM, Dobrescu T. Effect of Strength Training Protocol on Bone Mineral Density for Postmenopausal Women with Osteopenia/Osteoporosis Assessed by Dual-Energy X-ray Absorptiometry (DEXA). Sensors (Basel). 2022;22(5):1904. Published 2022 Feb 28. doi:10.3390/s22051904
- de Sire A, Lippi L, Venetis K, et al. Efficacy of Antiresorptive Drugs on Bone Mineral Density in Post-Menopausal Women With Early Breast Cancer Receiving Adjuvant Aromatase Inhibitors: A Systematic Review of Randomized Controlled Trials. Front Oncol. 2022;11:829875. Published 2022 Jan 21. doi:10.3389/fonc.2021.829875
- Yang L, Chen C, Zhang Z, Wei X. Diagnosis of Bone Mineral Density Based on Backscattering Resonance Phenomenon Using Coregistered Functional Laser Photoacoustic and Ultrasonic Probes. Sensors (Basel). 2021 Dec 9;21(24):8243. doi: 10.3390/s21248243.
- ⎫ The manuscript is revised based on this comment. Yes, the opinion of the honorable reviewer is absolutely correct. Based on this, we have added the requested content based on the references introduced to the manuscript and references 1, 12 and 13.
“Osteoporosis can be considered as a result of lack of calcium as well as other minerals in the bones. This disease can be classified into the following 3 categories: a) primary osteoporosis including idiopathic osteoporosis, type 1 osteoporosis due to estrogen deficiency common in postmenopausal women, type 2 osteoporosis due to estrogen deficiency with loss Bone density. Vertebral fractures can be considered as one of the common consequences of osteoporosis, which leads to constant pain, skeletal-muscular abnormalities, respiratory and deterioration of physical performance and quality of life. Also, there will be a chance of sustaining a fracture after an initial fracture. Physiotherapy can be considered a special non-pharmacological method, which can be useful and effective for patients who have osteoporosis, through manual methods, resistance exercises, etc.”
“Bone Mineral Density (BMD) is currently the main factor for determining bone quality and risk of osteoporosis. Dual Energy X-ray absorptiometry (DEXA) is considered the clinical gold standard for BMD evaluation. According to the guidelines of the National Osteoporosis Foundation, the indications for DEXA examination are: a) Women 65 years old and men 70 years and older. b) Postmenopausal women and men over 50 who have suffered bone fractures. c) Adults who suffer from rheumatoid arthritis or use medications such as glucocorticoids and have low bone mass.”
Which is highlighted introduction, pages 1-2 and lines 31-41 and 48-54.
- 2. METHODS: in my opinion, the whole section should be extensively reviewed. Indeed, reviewing the section rises several critical concerns, as it follows:
- (A) The sample size calculation should be better characterized, providing the rationale behind selecting the specific number of subjects and imaging centers. It is essential to provide a clear justification for the chosen sample size, considering the statistical power and representativeness of the study population.
- ⎫ The According to the opinion of the respected reviewer, the sample size is presented in accordance with the sample size in previous studies.
- (B) The 3-year period of investigation should be better clarified.
- ⎫ The manuscript is revised based on this comment. Yes, the opinion of the honorable reviewer is absolutely correct. The reason why it took 3 years to collect data is because of the Covid-19 epidemic. During the corona epidemic, the number of referrals for MRI scan has been very low. For this reason, the number of samples registered for MRA during 3 years was about 300. According to the opinion of the respected referee, the sentence of 3 years for the collection of images has been removed from the manuscript.
- (C) The study lacks critical information about the inclusion and exclusion criteria for the population assessed. Indeed, this is a crucial element which could help preventing sources of bias (e.g., selection bias).
- ⎫ The manuscript is revised based on this comment. The point to remember when recording MRI scans is that we did not take into account the number of images obtained based on the number of participants invited to the test. Patients with various diseases were referred to the imaging center by their doctors, and the purpose of the test and the expected results were explained to the patients during the patient admission process. We used the MRI scans taken from them in our research, and as an incentive gift due to the subjects' consent, a free DEXA image was obtained from each patient after taking the MRI scans, as well as information about the participants' BMD and bone condition. The necessary information was provided to the participants for free in the experiment based on the DEXA images obtained.
- ⎫ People who went to the imaging center with a doctor's referral and did not consent to use their images in this research after explaining the project, were left out of the testing process.
- ⎫ In general, these patients are excluded from the testing process: lack of personal consent for testing, people who used a pacemaker, people who had an implant or a metal foreign body in their body (such as having a metal piece in the brain, eye or platinum in bone).
Which is highlighted in section 2-1, page 2 and lines 95-127.
- (D) The demographic characteristics of the participants should be implemented. You might consider including further information (e.g., age, sex, BMI, comorbidities, etc.) within an additional table.
- ⎫ The manuscript is revised based on this comment. According to the opinion of the honorable reviewer, the characteristics of people in terms of gender and age are as follows:
Our collected data included 284 images taken from 21 to 83 years old Iranians with no history of bone fracture (135 men, average age of 56, standard deviation of 13, and with an average BMI of 27.8 and 149 women, average age of 49.5, standard deviation of 16 and with an average BMI of 24.3).
- ⎫ It is obvious that due to the lack of permission from the ethics committee, we are not able to present the details of each member participating in the experiment separately.
Which is highlighted in section 2-1, page 2 and lines 95-127.
- (E) DISCUSSION: while the results demonstrate promising accuracy rates, it is important to discuss the limitations of the study. Keeping in mind the aforementioned suggestions, I think that the section should be implemented addressing potential sources of bias, such as variations in MRI scan parameters. This would enhance the generalizability of your proposed algorithm. Moreover, further discussion on the clinical implications and potential applications of your proposed algorithm should be provided. Indeed, considering the ongoing advancements in the field, it would be beneficial to discuss your findings in the context of existing literature.
- ⎫ The manuscript is revised based on this comment. Yes, the opinion of the honorable reviewer is absolutely correct. Based on the reviewer's opinion, we have provided more discussion about our study and we have presented the limitations of our study in this section.
- ⎫ Despite its efficiency and effectiveness, this study has the following limitations:
First, when the number of databases grows, the current method will be unable to fulfill the needs of the medical community on a large scale, because the use of engineering features based on machine learning techniques does not ensure the optimality of the feature vector in diverse scenarios. Second, if the database is expanded, deep learning techniques should be applied and analyzed. Third, in order to increase data, the performance of generative adversarial network should be examined in future studies.
Which is highlighted in Discussion section, page 9 and lines 422-428.
- (F) ETHICS STATEMENT: please, provide the manuscript’s IRB.
- ⎫ The manuscript is revised based on this comment. We have provided the code of ethics along with the relevant license.
“We obtained informed consent of all subjects in writing prior to the tests and the tests were approved by the Ethics Committee of the hospital (with license number IRB.1398.9.26).”
Which is highlighted in section 2-1, page 2 and lines 95-127.
- (G) WHOLE MANUSCRIPT: please, note that numbers at the beginning of a sentence should be spelled (e.g., lines 14, 17, 113, etc.). Otherwise, you might consider starting the sentence with a connector.
- ⎫ The manuscript is revised based on this comment. Yes, the opinion of the honorable reviewer is absolutely correct. We have included the numbers throughout the manuscript along with a connector.
- (H) INTRODUCTION: lines 57-59. Apparently, the sentence should be changed as it follows: “[…] was practical or not”.
- ⎫ The manuscript is revised based on this comment. We corrected the sentence according to the opinion of the honorable reviewer.

Round 2
Reviewer 2 Report
Thanks for addressing my comments. I have one more comment about the revised manuscript.
In Fig. 6 and 7, the authors need to use the spatial resolution of each image to convert the values in pixel to millimeter, instead of simply changing the label in the figure. Additionally, please double check this will not affect the statistic functions of these curves.
Author Response
Reviewer#2:
Comments:
In Fig. 6 and 7, the authors need to use the spatial resolution of each image to convert the values in pixel to millimeter, instead of simply changing the label in the figure. Additionally, please double check this will not affect the statistic functions of these curves.
We rechecked Figures 6 and 7. For the opinion of the respected judge, the meaning in the figures is pixels and not millimeters. Accordingly, we have changed millimeters to pixels in Figures 6 and 7.

Reviewer 4 Report
Dear Authors,
in my opinion, the manuscript is interesting, and the results are intriguing.
You have significantly improved the paper during the revision process.
Best regards
Author Response
Reviewer#1:
Comments:
Dear Authors, in my opinion, the manuscript is interesting, and the results are intriguing.
You have significantly improved the paper during the revision process.
Best regards
- ⎫ Thank you for the careful opinion of the respected referee, we believe that your comments have helped to improve the scientific framework and writing of the article. Accordingly, we appreciate you.
With respect.
